# Immunopathology of Chronic Hepatitis B Infection: Role of Innate and Adaptive Immune Response in Disease Progression

**DOI:** 10.3390/ijms22115497

**Published:** 2021-05-23

**Authors:** Arshi Khanam, Joel V. Chua, Shyam Kottilil

**Affiliations:** Division of Clinical Care and Research, Institute of Human Virology, University of Maryland School of Medicine, Baltimore, MD 21201, USA; akhanam@ihv.umaryland.edu (A.K.); jchua@ihv.umaryland.edu (J.V.C.)

**Keywords:** CHB, innate immune cells, adaptive immune cells, inflammation

## Abstract

More than 250 million people are living with chronic hepatitis B despite the availability of highly effective vaccines and oral antivirals. Although innate and adaptive immune cells play crucial roles in controlling hepatitis B virus (HBV) infection, they are also accountable for inflammation and subsequently cause liver pathologies. During the initial phase of HBV infection, innate immunity is triggered leading to antiviral cytokines production, followed by activation and intrahepatic recruitment of the adaptive immune system resulting in successful virus elimination. In chronic HBV infection, significant alterations in both innate and adaptive immunity including expansion of regulatory cells, overexpression of co-inhibitory receptors, presence of abundant inflammatory mediators, and modifications in immune cell derived exosome release and function occurs, which overpower antiviral response leading to persistent viral infection and subsequent immune pathologies associated with disease progression towards fibrosis, cirrhosis, and hepatocellular carcinoma. In this review, we discuss the current knowledge of innate and adaptive immune cells transformations that are associated with immunopathogenesis and disease outcome in CHB patients.

## 1. Introduction

Despite the availability of highly effective preventive vaccines and oral antivirals, an estimated 250 million people are chronically infected with hepatitis B virus (HBV) [1]. HBV infection is one of the leading causes of cancer related death worldwide. Most people develop acute self-limiting infection that get clear with strong host immune response; however, those who do not clear develop chronic infection, which progress towards fibrosis, cirrhosis, and hepatocellular carcinoma (HCC) leading to high mortality [2].

The outcome of chronic HBV infection is determined by virus–host interactions. HBV itself is a non-cytopathic virus and liver damage is mainly attributed to the host immune response. During HBV infection, host immune response acts as a double-edged sword; it provides defense towards infection by destroying the virus infected cells, whilst induce hepatic inflammation and aggravate liver injury. Although host immunity constitute different cell types, CD8 T cells are considered as the major factor responsible for hepatic damage during acute HBV infection [3]. HBV-specific CD8 T cells directly attack infected hepatocytes and subsequently recruit other components of the immune system, causing immunopathogenesis and further hepatic damage [3]. However, in chronic infection, HBV-specific CD8 T cells acquire exhaustive phenotype and produce less inflammatory cytokines [4], indicating that HBV-specific CD8 T cells might not be a major mediator of liver injury, instead liver injury is driven by the intrahepatic recruitment of other immune cells.

Since chronic HBV infection is diagnosed after several weeks or months of infection when the virus is already escaped and viremia is high, adaptive immune response is appraised for efficient viral control and innate immune cells are overlooked. However, both innate and adaptive immune responses have important and diverse functions during HBV infection. Antigen presenting cells (APCs) recognize different viral proteins and viral nucleic acids through pattern recognition receptors (PPRs) including toll-like receptors (TLRs), resulting in rapid antiviral cytokine production and other immune cells activation leading to early control of HBV infection [5,6,7]. Moreover, activation of the innate immune pathways mediates the recruitment of adaptive immune cells, which then perform HBV-specific functions by specifically recognizing the virus infected hepatocytes and killing them [8,9,10]. Subsequently, these cells develop HBV-specific memory, which protects from future HBV infection. The role of classical CD4 and CD8 T cells have been studied for decades and cell mediated immunity was found to be crucial for HBV clearance. Both CD4 and CD8 T cells work synergistically to control HBV infection. In HBV infected chimpanzees, CD4 and CD8 T cells helped in resolution of infection by producing interferon-γ (IFN-γ) and tumor necrosis factor- α (TNF-α) cytokines [11]. Depletion of CD4 caused reduced CD8 T cell response during acute infection, while deficiency of CD8 T cells resulted in failure of HBV clearance in chronic infection, suggesting a crucial role in viral elimination.

In this review, we will discuss current knowledge on innate and adaptive immune response and their association with immunopathogenesis of HBV infection.

## 2. Innate and Adaptive Immune Response against HBV Infection

Innate immune response is important in the early management of HBV infection and limits the disease at initial stage; later, it helps in generating a proficient adaptive immune response that clears the infection. PRRs recognize different viral components including envelope proteins, nucleocapsid, nucleic acids, and specific viral structures that activate immune cells and signaling pathways to encourage the production of pro-inflammatory cytokines, chemokines, and interferons [12,13]. Both circulating as well as the intrahepatic innate immune system can sense and respond to HBV infection. However, robust immune response also leads to hepatic necro-inflammation causing severe liver damage. Numerous innate immune cells including dendritic cells (DCs), macrophages, monocytes, natural killer (NK) cells, myeloid derived suppressor cells (MDSCs), and innate lymphoid cells (ILCs) play a protective as well as pathogenic role during chronic HBV infection. Immediately after a pathogen encounter, activation of the innate immune system occurs that is necessary for the recruitment and activation of adaptive immunity [14]. Adaptive immune systems act through the expansion and functional maturation of discrete T and B cell subsets that particularly recognize and kill HBV infected hepatocytes; a process that induces hepatic inflammation. Persistent exposure of viral factors, including hepatitis B surface antigen (HBsAg), hepatitis B e antigen (HBeAg), and hepatitis B x antigen (HBx), leads to immune exhaustion and subsequent downregulation of host response by setting up chronic infection [15]. HBsAg is present on the surface of HBV and is responsible for binding and entry into hepatocytes. HBsAg can be detected in the blood after several weeks of infection and its presence indicates that the patient has contracted the infection. Production of anti-HBs antibodies is critical for viral clearance, long term protection, and defines functional cure [1]; whereas, HBeAg is regarded as an accessory protein as it is not required for the viral genome replication. It can be found between icosahedral nucleocapsid core and the lipid envelope (the outermost layer of HBV) and its existence designates active viral replication. HBeAg exerts its immunoregulatory effect by eliciting tolerance in hepatitis B core antigen (HBcAg)/HBeAg-specific T cells [5]. HBcAg is located at the surface of the nucleocapsid core (the innermost layer of the HBV). Presence of anti-HBc antibodies reflect past or current HBV infection and these antibodies appear within a few days of infection; however, do not provide any protection against HBV, unlike the surface antibody [1]. Moreover, HBxAg stimulates virus gene expression and replication and is crucial for the establishment and maintenance of chronic carrier state. Intrahepatic inflammatory reactions [16,17,18] lead to the induction of several suppressive pathways and subsequent recruitment of regulatory cells that drive functional demolition of T cells [19,20,21,22,23]. These cells start overexpressing inhibitory receptors [9,24,25,26,27] that further dampen their functional status causing immune exhaustion, resulting in viral persistence and further disease progression [28]. Concisely, both innate and adaptive immune systems work synergistically to cause immune related pathologies in CHB (Figure 1).

### 2.1. Role of Innate and Adaptive Regulatory Cells in CHB

#### 2.1.1. Regulatory Dendritic Cells

Generally, regulatory cells are involved in the modulation of other immune cells by promoting self-tolerance and suppression of non-self-immune response. During chronic HBV infection, presence of several innate and adaptive regulatory cells has been reported [23,29]. DCs, one of the most potent APCs, play a crucial role in the initiation and maintenance of T cell functions. Functional defects in DCs derive T cell immune tolerance to viral infections. DCs are not a single cell type but rather a heterogeneous population of cells that have developmental plasticity. Different factors including ligands for PPRs and cytokines, alters DCs function [30]. Treatment of DCs with immunosuppressive molecules leads to the generation of tolerogenic DCs that down regulate major histocompatibility (MHC) complex and costimulatory molecules, drive defective T cell activation, and stimulate other regulatory cells as well as anergic T cells [31,32,33]. Regulatory dendritic cells (DCregs) are a recently identified subset of DCs that possess regulatory function and are characterized by a reasonably low expression of costimulatory molecules and MHC complex. DCregs have different cytokine pattern and perform their regulatory function through the induction of Tregs as well as T cell anergy and depletion. Although DCregs are important for the downregulation of exaggerated immune response and inflammation, they also inhibit effector T cell functions, required for viral clearance. As DCregs are a newly identified subset, there is paucity of data in CHB. One study reported that HBeAg negatively affects the generation of DCs [34]. During chronic HBV infection, DCs acquire regulatory phenotype in response to HBV-specific stimuli, particularly HBeAg and assist in viral immune escape. DCs incubated with HBeAg were defective in IL-12 production and further T cell proliferation [19]. This supports the concept that HBeAg can condition innate immune cells into anti-inflammatory phenotypes. Both HBeAg and HBcAg upregulates B7-H1 expression on DCs, causing functional impairment and further viral persistence [35]. Binding of B7-H1 to programmed death-1 (PD-1) delivers co-inhibitory signals to T cells, which regulate T cell function and generate immune tolerance. 

#### 2.1.2. Myeloid Derived Suppressor Cells

Myeloid derived suppressor cells (MDSCs) are regulatory cells, originated from myeloid progenitors and expand under different pathological conditions. These cells are a mixed population of monocytic-MDSCs (M-MDCs) and granulocytic-MDSCs (G-MDCs). Although MDSCs are associated with protective function in acute liver inflammation, these cells facilitate inflammation and tissue damage during chronic liver disease. The expansion of MDSCs has been reported during chronic HBV infection, which favors T cell exhaustion and establishment of persistent HBV infection [36,37]. The frequencies of circulating MDSCs were higher in HBeAg-positive patients than HBeAg-negative. HBeAg stimulates MDSCs expansion through the upregulation of indoleamine-2, 3-dioxygenase (IDO) that plays a major role in suppressing T cell proliferation [21]. Moreover, MDSCs purified from HBeAg-positive patients reduced the proliferation of both CD4 and CD8 T cells that were restored after IDO neutralization, suggesting the involvement of MDSCs in the development of HBeAg-induced immune tolerance. Persistence of these cells shas been demonstrated despite viral load reduction after the antiviral treatment. Enhanced HLA-DR^−^CD11b^+^CD33^hi^ M-MDCs correlate with increased HBsAg concentration and Tregs. Both M-MDSCs and G-MDSCs produce high amount of TGF-β and IL-10 that promote the development of induced Tregs (iTregs) [22]. Even one year of Tenofovir treatment failed to normalize the frequencies, function, and reduction in Tregs percentage. The frequencies of MDSCs correlate with serum HBV viral load. High number of MDSCs is proficient in suppressing HBV-specific T cell response including IFN-γ, granzyme B and perforin production, and CD107a expression. These cells alter T cell function by several mechanisms including PD-L1 expression, reactive oxygen species (ROS) and IDO secretion, production of arginase1 (Arg1), upregulation of nitric oxide synthase 2 (iNOS2), and prostaglandin E2 (PGE2). HBsAg induces MDSCs expansion via ERK/IL-6/STAT-3 feedback signaling [38]. Previous data reported that M-MDSCs suppress T cell activation through PD-L1, while G-MDSCs develop immune suppression via Arg1 expression in persistent HBV infection [39]. Both Arg1 and iNOS2 disrupt IL-2 signaling and inhibit T cell proliferation.

MDSCs secrete immunosuppressive cytokines including TGF-β and IL-10. TGF-β suppresses NK cell function by decreasing IFN-γ production and inhibiting B cell proliferation [40]. TGF-β is also responsible for tumor progression by inhibiting anti-tumor activity of T cells [41]. Immunosuppressive roles of MDSCs are mainly directed towards T cells, although some studies suggest that they also act in the regulation of DCs, B cell, and macrophage-mediated response. 

#### 2.1.3. Regulatory T Cells

Regulatory T cells (Tregs) exhibit immunosuppressive function and play a critical role in the development and maintenance of immunological tolerance by suppressing various cell types including DCs, NK and NKT cells, CD4, and CD8 T cells [42]. Tregs also limit antigen-specific immune response and anti-tumor immunity. They perform their function by secreting immunosuppressive mediators, such as IL-10 and TGF-β, and through contact-dependent mechanisms. Presence of co-inhibitory receptors including cytotoxic T lymphocyte antigen-4 (CTLA-4) and T cell immunoreceptor with Ig and ITIM domains (TIGIT) on Tregs support immune suppression [43]. These cells suppress IL-2 production that in turn inhibits T cell proliferation. The frequencies, phenotypes and function of Tregs varies between circulation and intrahepatic compartment [44]. Intrahepatic Tregs hinders the function of effector T cells, establish immunosuppressive microenvironment, and are accountable for disease progression. Broadly, Tregs are classified into natural as well as induced Tregs. Natural Tregs originate in the thymus after ligation of high affinity T cell receptors, whereas induced Tregs generate from naïve CD4 T cell precursors in the periphery. Normally, Tregs relieve liver inflammation and immune mediated liver injury; however, they may also encourage apoptosis-induced inflammation.

During acute HBV infection, Tregs protect the liver from immune mediated liver damage [44,45]; while in chronic HBV infection, Tregs are involved in the development of cirrhosis and further transformation to HCC and metastasis, suggesting different roles of Tregs in various disease stages [44]. CHB patients display higher frequencies of Tregs in the liver as well as circulation than that of asymptomatic HBV-infected patients, inactive HBsAg carriers, and acute HBV infected patients [44]. Increased infiltration of Tregs in the tumor of HCC patients with HBV pre-S2 mutant has been reported, suggesting that increased Tregs are associated with chronicity of HBV infection and occurrence of high Tregs may define disease severity [46]. HBeAg-positive patients display higher Tregs than HBeAg-negative patients [47]. HBeAg stimulates TGF-β secretion resulting in expansion of Tregs by transforming CD4+CD25- T cells into CD4+CD25+Foxp3+ Tregs [48]. Hepatitis B induces immune cells to establish TGF-β rich microenvironment for Treg differentiation favoring HBV persistence. Moreover, HBeAg encourages NKG2A+ NK cell dysfunction through Tregs derived IL-10. Blockade of IL-10 leads to the reduction of NKG2A+ NK cells and increases IFN-γ+ NK cells [49].

#### 2.1.4. Regulatory B Cells

B cells are known for their antibody production and differentiation into plasma cells providing long-term immunity. Emerging evidence reveal that B cells have other modulatory functions that are not involved in antibody production; these cells have been defined as B regulatory cells (Bregs). Bregs are different subset than effector B cells and possess different cytokine profile [50]. These cells secrete IL-10, which serve as a crucial mediator for B cell-derive regulation of other immune cells, and maintenance of immune tolerance. IL-10 producing B cells are augmented among CD19^+^CD24^hi^CD38^hi^ transitional B cells and CD19^+^CD24^hi^CD27^+^ B cells or both cell types. CD19+CD24^hi^CD38^hi^ B cells constitute a subset of immature B cells that have been termed as transitional/regulatory B cells. These cells play a crucial role in regulating T cell responses by releasing IL-10. In vitro stimulation of these cells with CD40 induces IL-10 secretion and inhibits IFN-γ and TNF-α production by CD3+CD4+CD25- T cells [51]. Moreover, these cells inhibit the differentiation of Th1 and Th17 cells from naïve T cells. Similarly, CD19^+^CD24^hi^CD27^+^ B cells also define regulatory B cells that suppress T cell activity by secreting IL-10 [52]. Recently, the role of Bregs has drawn much attention during chronic viral infection. Bregs exert several negative regulatory functions on the effector immune cells and contribute to the impairment of HBV clearance. CHB patients have high Bregs that were higher in the immune active group than the immune tolerant and healthy controls [20,53]. An increased Bregs population can promote HBV replication and liver fibrosis [54], whereas it can also decrease alanine Aminotransferase (ALT) levels by reducing the liver inflammation. Bregs employ regulatory functions mainly by producing IL-10 and TGF-β that inhibit the production of pro-inflammatory cytokines by CD4 cells [55], and support the differentiation of CD4+ CD25- T cells into Tregs, and further inhibit HBV-specific T cell response [52]. Blockade of IL-10 inhibited the regulatory effects of Bregs on CD4 T cells T cells and abrogated the conversion of Tregs [56]. Moreover, HBV-specific CD8 T cell responses were rescued after IL-10 neutralization. Increased Bregs are found in CHB patients with spontaneous flare, reflected by increased viral load and liver inflammation, suggesting that Bregs are accountable for HBV flare by suppressing HBV-specific CD8 T cells and restricting viral control [57]. Moreover, Bregs associate with advanced histological fibrosis stage and enhances HBV replication. Co-culture of Bregs with CD4+CD25- T cells suppress IFN-γ and IL-17 production and enhance IL-4 secretion. Additionally, depletion of Bregs decreased Treg numbers and expression of co-inhibitory molecule CTLA-4, reduced IL-10 and TGF-β secretion [51]. Elevated Bregs can overpower effector T cells and enhance Tregs that generate immune tolerance in chronic HBV infection. Altogether, both innate and adaptive regulatory cells play a critical role in the immunopathogenesis of CHB patients mainly through the production of IL-10 and TGF-β.

### 2.2. Involvement of Innate and Adaptive Inhibitory Receptors in CHB

#### 2.2.1. Inhibitory Receptors on NK Cells

Hepatitis B infection alters the surface receptor expression and activation status of innate and adaptive immune cells. During CHB, NK cells display higher expression of inhibitory receptor NKG2A, while activation receptors CD16 and NKp30 were downregulated [58,59]. These changes are associated with serum HBV DNA load. Moreover, classical NK cell receptors and co-inhibitory receptors may impair NK cell function. Increased T cell immunoglobulin and mucin-domain containing-3 (TIM-3) expression impair NK function and in vitro blockade of TIM-3 improved NK cell function [60]. NK cells also express PD-1, TIGIT, CD96, CD266, and IL-1R8 that regulate NK cell function. Inhibition of IL-1R8 restore NK cell maturation and effector function while binding of TIGIT and CD96 to CD266 counterbalance NK cell function. NKG2A has served as an important checkpoint to recover hepatic NK cell function. Reduced NK cell activation and function is also related to the ligand expression for inhibitory and activating NK cell receptors. NK cells express TNF-related apoptosis inducing ligand (TRAIL) that induce T cell death. TRAIL-positive NK cells mediate the apoptosis of HBV-specific CD8 T cells by binding to TRAIL-R2 receptor present on CD8 T cells [61]. Blockade of TRAIL in in vitro condition enhances HBV-specific CD8 T cells. As T cell apoptosis is dependent on NKG2D and TRAIL, activated T cells are highly sensitive for killing due to high expression of NKG2D and TRAIL on NK cells during chronic HBV infection. However, NK cell activating receptor 1 NKp46 can efficiently control TRAIL expression. Therefore, these findings suggest NK cells as potential regulators of antiviral T cell response. Blocking these inhibitory receptors might be a critical approach for the treatment of CHB.

#### 2.2.2. Inhibitory Receptors on CD4 and CD8 T Cells

During chronic HBV infection, broad acting and virus-specific T cells over express co-inhibitory molecules [62,63,64]. While overexpression of inhibitory receptors is intended to control immune hyperactivation and escape immune pathology, constant overexpression of these molecules leads to functional exhaustion of T cells [65]. These exhausted T cells correlate with a hierarchical dysfunction of their proliferative abilities and effector function and are prone to increased apoptosis. During acute HBV infection, effector T cells retain higher PD-1 expression that tend to decline in the recovery phase [66]. PD-1 obstructs T cell receptor mediated signaling and induces functional exhaustion in T cells by inhibiting effector responses including cytokine production and cytolytic activities in CHB patients [37,67]. Blockade of PD1 with anti-PD-1 antibody partially recovers T and B cell responses in CHB patients, which serve them as potential therapeutic target for the treatment of chronic HBV infection [62,68]. Continuous antigen exposure maintains high inhibitory receptors expression resulting in impaired T cell function [69]. T cells express several inhibitory receptors including CTLA-4, 2B4, LAG3, CD160, TIM-3, and galectin-9. Liver-infiltrating HBV-specific T cells display highest PD-1 expression among several other inhibitory receptors and their corresponding ligands are expressed by APCs [70]. Ligation of PD-1 with its ligand inhibit the activation of costimulatory receptors as well as other TCR components and tempers the analogues signaling pathway, further upregulating inhibitory genes.

Blockade of inhibitory checkpoints including PD-1, CTLA-4, 2B4, TIM-3, and galectin-9 alone or in combination has emerged as a potential therapeutic approach to restore T and B cell functions in CHB [27,63,64,70,71,72]. Previous data revealed that PD-1 pathway is critical for CD8 T cell exhaustion and blockade of PD-1 restores the function of HBV-specific CD8 T cells [73]. However, the exhaustion could not be completely recovered by PD-1 blockade alone and complete functional restoration requires a combined PD-1/CTLA-4 inhibition. Constant HBsAg and HBeAg exposure exhaust a large quantity of CD8 T cells which gradually upregulates CTLA-4 expression [27,73]. Recently, it has been reported that CTLA-4 expression on Tregs inhibit the function of T follicular helper (T_FH_) cells and its blockade with CTLA-4 neutralizing antibody restore the ability of T_FH_ cells to clear the infection [27]. T_FH_ cells are mostly associated with B cell response. They are vital for the development of germinal centers from which high-affinity memory B and long-lived plasma cells are generated, which are required for protective antibody response [74]. HBeAg increased CTLA-4 expression on CD8 T cells and that was associated with high HBV DNA load [73]. Similarly, CD4 T cell proliferation is affected by the upregulation of CTLA-4. CD4 T cells are polarized towards Th2 or inducible T regulatory phenotype, increasing the levels of anti-inflammatory cytokines. Increased expression of TIM-3 and galectin-9 contribute to the functional inhibition and apoptotic deletion of T cells [64]. T cells expressing TIM-3 are defective in producing IFN-γ and TNF-α upon recognition of HBV peptides and are vulnerable to galectin-9 triggered cell death. Additionally, frequency of TIM-3 expressing T cells negatively correlates with T-bet (T-box expressed in T cells) messenger RNA (mRNA) expression and plasma IFN-γ levels. T-bet is a transcription factor, which plays an essential role in the differentiation of IFN-γ producing Th1 cells that are required for antiviral response. Expression of TIM-3 on peripheral T cells correlates with disease progression and markers of liver injury including increase in ALT, AST (aspartate aminotransferase), bilirubin, and international normalized ratio [75]. Blockade of TIM-3 signaling induced proliferation and expansion of HBV-specific CD8 T cells, enhanced antiviral cytokine secretion, and significantly reconstituted the HBV-specific CD8 T cell response [76]. HBV-specific CD8 T cells also possess higher expression of apoptosis gene, Bim, that induces apoptosis and contributes to exhausted CD8 T cells and obstructs their response leading to persistent viral infection [77].

#### 2.2.3. Inhibitory Receptors on B Cells

Several studies reported the presence of inhibitory receptors on B cells. RNA sequencing of B cells isolated from CHB patients demonstrated that B cells have upregulated expression of multiple inhibitory receptors including members of the Fc receptor-like (FcRL) family. Fc receptors regulate the antigen-driven activation and expansion of B cells. FCγRII is a potent inhibitor of B cell antigen receptor (BCR) signaling. It inhibits BCR signaling after binding to BCR along with antigen containing immune complexes. It is known that HBV protein and CD40L upregulates inhibitory receptors on B cells. HBcAg upregulates FcRL4, FcRL5, and PD-1 on B cells [78]. HBV-specific B cells retain FcRL4, FcRL5 and FCγRII, BTLA, CD22, and PD-1 and impair anti-HBs function as well as T_FH_-B cell axis [24]. Moreover, FcRL5 has been implicated as the key factory in the generation of atypical memory B cells (atMBCs). These cells associate with abnormal T_FH_ cell expansion expressing high CD40 ligand, which in turn, correlate with altered B cell differentiation and higher accumulation of atMBCs in CHB patients, resulting in persistent viral infection [78]. While atMBCs present in the liver under normal conditions, they expand under pathological microenvironment. Frequencies of atMBCs are higher in HBV-infected liver than peripheral blood. Accumulation of atMBCs inhibits antigen-specific B cell response and HBsAb production. Intrahepatic atMBCs possess higher PD-1 expression that further induce functional impairment in these cells [24] by inhibiting signal transduction, homing, survival, and differentiation into antibody producing cells and lead to B cell defects in CHB. Blockade of PD-1 with anti-PD-1 antibody partially improves dysfunctional virus-specific B cells, which suggests that anti-PD1 therapy in CHB might be able to recover both HBV-specific T and B cell functionality [68]. Inhibition of these receptors represents novel B cell therapeutic targets.

### 2.3. Involvement of B Cells in CHB

Growing evidence suggests the critical role of B cells in providing defense against chronic HBV infection by producing humoral antibodies [79]. B cells secrete antibodies against different viral proteins including HBs, HBcAg, and HBeAg [80]. During chronic HBV infection, the presence of both HBs and HBcAg-specific B cells have been reported. While HBcAg-specific B cells are present at higher frequencies and produce humoral antibodies, HBsAg-specific B cells are present in lower frequencies and are defective in antibody production [24,68]. HBcAg-specific B cells are favorably IgG+ memory B cells. Despite the phenotypic and functional differences between HBs and HBcAg-specific B cells, they share common mRNA expression patterns that vary from global memory B cells and are defined by high expression of genes related to cross presentation and innate immune activity [81]. Our recent study reported that stimulation of naïve and memory B cells with recombinant IL-27 partially recovers the HBsAg-specific protective antibody secretion in CHB patients by supporting the generation of plasmablasts and plasma cells and enhancing BLIMP-1 expression [20]. Ex vivo data also revealed the presence of higher plasmablasts and plasma cells in CHB patients. In CHB, B cells are majorly focused on antibody production; however, other functions including antigen presentation and immune regulation have been overlooked, which are associated with immune tolerance, HBV persistence and liver injury.

Existence of IgG antibody against HBcAg has been detected during prior, ongoing and even occult HBV infection, whereas, anti-HBc IgM antibody is present only during an acute HBV infection and serious worsening of chronic infection [82]. Anti-HBe antibodies emerge earlier than anti-HBc and could predict a better outcome. However, immune control of HBV infection requires robust HBsAg-specific humoral antibodies. Recent studies demonstrated HBsAg-specific humoral response in the circulation of CHB patients and found that HBsAg-specific B cells contain atMBCs cells expressing high PD-1 and blockade of PD-1 partially restored the function of HBsAg-specific B cells [68]. Moreover, addition of IL-2, IL-21, and CD40L is also useful for the partial recovery of HBsAg-specific B cell function [24,68,83]. Since anti-HBs neutralizing antibodies are critical to control viral infection, impaired anti-HBs antibody production leads to persistent HBV infection. B cell responses are primarily regulated by the T_FH_ cells mainly through IL-21. However, during CHB, T_FH_ cells are impaired in producing HBsAg-specific IL-21 that is mediated by Tregs and follicular regulatory T cells, while depletion of Tregs reinstates T_FH_ cell function [27]. Our data demonstrated that regardless of defective IL-21 secretion, T_FH_ cells support B cell function by secreting IL-27 that support plasmablasts and plasma cell formation and partly encourage HBsAg-specific IgG and IgM secretion [20]. Although reduced differentiation of antigen-specific memory B cells into anti-HBs secreting plasma cells affect HBV-specific humoral response in CHB patients leading to persistent HBV infection and, subsequently, liver damage.

During chronic HBV infection, B cells also serve as APCs. HBcAg-specific B cells present HBcAg to helper T cells more efficiently than non-B cell APCs, which is confirmed recently by examining transcriptomic profile of B cells [81]. During chronic HBV infection, several studies supported the role of B cells as APCs by assessing the risk of HBV reactivation after rituximab therapy (anti-CD20 antibody) in B cell lymphoma patients [84,85]. Rituximab induces HBV reactivation independently and in combination with chemotherapy. When combined with chemotherapy, HBV reactivation rate could be as high as 20–55% overall and 3% in HBsAg negative patients, confirming the critical role of B cells in providing protection against HBV infection [86]. One of the reasons for HBV reactivation during combined chemotherapy and rituximab is the decrease in antibody titers that is associated with B cell depletion. Chemotherapy induced reactivation is more dangerous than acute HBV infection, affecting the chances of survival; therefore, it is critical to control HBV reactivation. B cells express BCRs that provide specificity during the recognition and binding of HBV antigen. High affinity BCRs allow B cells to present specific antigens with high proficiency even in the presence of tremendously low antigen concentrations [87]. Interestingly, follicular B cells can identify HBV antigens that not only exist alone but are present on the surface of macrophages or DCs [88]. CHB patients displayed high B cell activating factor (BAFF) that is required for B cell activation. Moreover, monocytes also secrete BFF in response to HBeAg that might be associated with hyper B cell activation in CHB. The elevated BAFF levels are related with clinical outcomes, particularly liver cirrhosis and HCC [89]. Importantly, serum BAFF level is not only an independent variable associated with HCC but also has higher AUC value than AFP level serving it as a biomarker for the diagnosis of HBV related clinical diseases [90]. In summary, B cells are not only crucial in viral control but are associated with disease progression and liver damage in CHB.

### 2.4. Innate and Adaptive Immune Cell Derived Inflammatory Mediators in CHB

Cytokines are intracellular mediators secreted by both innate and adaptive immune cells are involved in viral control. After HBV infection, activation of different immune cell triggers the complex cytokine cascade and further generates the protective immune response. Cytokines are necessary for cellular activation, intracellular signaling, and cell–cell communication. They provide defense against different pathogens. During chronic HBV infection, cytokines play a critical role in immune regulation and inflammation. They inhibit viral replication; however, also influence the persistence of HBV infection. They induce the differentiation and maturation of different immune subsets into specialized effector cells having inimitable skills to protect against specific types of infection. They are also involved in the pathogenesis of CHB and its progression towards cirrhosis and HCC. Different factors including inflammation, fibrosis, viral load, and the occurrence of malignancy influence the cytokines levels. Discrete cytokine patterns in CHB patients with and without HCC, propose a significant role in immunopathogenesis.

Interferons α, β, and γ have an essential role in escalating innate immune response against CHB. Chronic HBV infection suppresses the production of IFN-α/β/γ and further temper cellular responses to IFN, affecting the activation of other cellular pathways and mechanisms [91]. IFN-α/β produced by plasmacytoid DCs and IFN-γ produced by macrophages, NK, NKT cells, and T cells generate systemic antiviral response. However, the aggravation of antigen non-specific cytokine response mediates hepatic inflammation and subsequently liver damage. Kupffer cells (KCs) produce chemokines CXCL9 and CXCL10 in HBV transgenic mouse models that mediate the infiltration of several inflammatory plasmacytoid cells to the liver and induce hepatic inflammation [92]. Several lines of evidence suggest an elevated level of IL-10 in CHB. Both HBsAg and HBeAg induce IL-10 production by peripheral blood mononuclear cells, mainly different regulatory cells, T cells, and monocytes, which develop immune tolerance against HBV, and advances fibrosis development [29,56,93,94]. NK cells mediate liver damage by producing IL-8/CXCL8 that enhances the expression of apoptosis receptors on hepatocytes [95]. Moreover, IL-10 produced by Th2 cells inhibits monocyte, macrophage, and T cell function by preventing IL-1, IL-6, IL-8, IL-12, and TNF-α production [96]. Similarly, different cytokines have diverse roles during CHB, which we have illustrated in Table 1.

### 2.5. Function of Innate and Adaptive Immune Cell Derived Exosomes in CHB

Exosomes are a type of extracellular vesicles that act as a carrier for the transportation of biologically active molecules between different cells [131,132]. Immune cells including DCs, macrophages, mast cells, T cells, and B cells release exosomes under normal condition as well as various pathological environments [133]. The amount of exosomes secreted may vary between normal and pathological settings [134]. Exosomes regulate cellular microenvironment, genetic and epigenetic mechanisms, and are involved in immunopathology [135]. During viral infection, exosomes contribute to immune regulation, antiviral response, as well as spread of the disease. Emerging data revealed that exosomes act as carriers for HBV particles and contribute to viral replication and pathogenesis. They carry viral genome and protein components, shuttle them from infected to uninfected cells, and facilitate the spread of HBV. During CHB, an increase in exosomal microRNAs miR-21 and miR-29 inhibit the release of IL-12 from DCs and macrophages, which inhibit NK cell activation and subsequent immune responses resulting in disease progression towards fibrosis [136]. Recently, it has been recognized that exosomes contain HBV particles and infect normal hepatocytes and NK cells resulting in NK cell dysfunction [137]. It is already established that NK cell functions are impaired during persistent HBV infection exhibiting lower IFN-γ secretion and cytolytic potential; hence, it is acceptable that exosomes are one of the major contributors for impaired NK cell function and innate immune regulation during chronic HBV infection. THP-1 (a human monocytic cell line that morphologically resembles with monocytes and macrophages) derived macrophage exosomes target hepatocytes through TIM-3 and subsequently deliver IFN-α induced anti-HBV activity [138]. Furthermore, macrophage derived exosomes facilitate antiviral activity during chronic HBV infection through miR-574-5p. Exosomes treated with pegylated interferon-α (PegIFN-α) exhibited anti-HBV activities including the suppression of HBsAg, HBeAg, HBV DNA, and covalently closed circular DNA (cccDNA) levels in HBV cell lines, suggesting exosomes can transport IFN-α related miRNAs from macrophages to HBV-infected hepatocytes [139]. Treatment with PegIFN-α upregulated exosomal hsa-mir-193a-5p, hsa-miR-25-5p, and hsa-miR-574-5p that partially inhibited HBV replication and transcription. Hsa-miR-574-5p leads to decline in pregenomic RNA and polymerase mRNA levels by binding to the 2750–2757 position of the HBV genomic sequence. Activated T cells also release exosomes and possess the proteins present in several other exosomes such as CD63, CD81, annexins, heat shock proteins, and enolase. Moreover, they contain β2 microglobulin, components of the TCR/CD3 complex, and specific integrins, among several others. CD4+ T cell derived exosomes have been shown to inhibit CD8 T cell response and antitumor immunity [140]. However, T cell derived exosomes have been rarely studied in HBV infection.

## 3. Involvement of Innate and Adaptive Immune Cells in the Development of Fibrosis, Cirrhosis, and HCC

CHB patients are at increased risk of developing fibrosis, cirrhosis, and HCC. Several lines of evidence suggest that HBV itself is non-cytopathic for infected hepatocytes and the liver injury is primarily dictated by the extravagant host immune response. During chronic HBV infection, continuous viral exposure induces the activation of different hepatic immune cells; these cells secrete abundant pro-inflammatory and fibrogenic factors, provoking hepatic inflammation and subsequently fibrosis [141]. Among several immune cells, cytotoxic T lymphocytes were considered as the main culprit for hepatic damage during acute HBV infection [142]. These cells directly kill the infected hepatocytes, which contributes to liver pathogenesis. Nevertheless, the exhaustion of HBV-specific CD8 T cells in chronic HBV infection reveal that these cells are not the only major contributor of liver injury during chronic HBV infection [143]. Rather, infiltration of other mononuclear cells including monocytes and macrophages is also involved. NK cells participate in the inflammatory process even before the intrahepatic recruitment of CTLs. Later on, HBV-specific CTLs induce sufficient toxicity and are involved in viral pathogenesis [144]. Similarly, other immune cells are also linked with the pathogenesis of chronic HBV infection.

### 3.1. Monocytes/Macrophages

Monocytes and macrophages play various roles during chronic HBV infection. HBV induces the production of TGF-β and IL-10 by monocytes/macrophages and inhibits TNF-α secretion [28]. Quantification of serum cytokines in HBV infected patients during the pre-symptomatic phase reveal that HBV infection did not stimulate the production of IFNs and IL-15, rather it induced IL-10 production [145]. TGF-β is one of the most profibrogenic cytokines involved in hepatic stellate cell (HSC) activation, fibrosis, and cirrhosis development. Binding of TGF-β to its receptor leads to the induction of myofibroblasts and matrix deposition. TGF-β stimulates quiescent HSCs trans-differentiation into myofibroblasts that secrete ECM [146,147]. A study performed in HBV humanized mice model revealed that HBV induces human monocytes/macrophage differentiation into M2 macrophage phenotype, an anti-inflammatory phenotype, expressing IL-10 and TGF-β along with other suppressive cytokines [148]. Similarly, in CHB patients, monocytes produce more IL-10 and TGF-β and express high inhibitory molecule, PD-L1. In vitro, studies revealed that HBsAg and HBV DNA directly induce PD-L1 expression and anti-inflammatory cytokines from monocytes of healthy individuals [149]. Moreover, hyperactivated pro-inflammatory CD16+ monocytes are associated with the severity of liver injury and fibrosis. CD16+ monocytes favorably release inflammatory cytokines and induce T helper 17 (Th17) cell expansion, a critical mediator of hepatic inflammation [150]. These cells produce IL-17 cytokine in the liver and initiate the mobilization and recruitment of activated neutrophils, driving extensive tissue inflammation and disease progression. In addition, hepatic macrophages promote HCC development by producing TNF-α and IL-6 [151]. CCL18^+^ M2 macrophages present in the tumors of advanced HCC promote angiogenesis, tumor invasion, metastasis, and associate with poor prognosis [152]. Hence, monocytes and macrophages play a key role in the immunopathogenesis of HBV infection.

### 3.2. NK Cells

NK cells present one of the major innate cell types in the liver. Increased frequencies and function of NK cells have been reported during chronic HBV infection. These cells highly express activating receptors and efficiently kill the infected cells, shown by hepatocyte and stellate cells lysis [153,154]. In CHB, increased expression of intrahepatic TRAIL, a death ligand, is associated with liver damage by binding to its receptor present on the surface of hepatocytes [155]. Moreover, NK cells from HBV-related inflammation induce apoptosis of primary hepatocytes. DC-activated NK cells are also capable of inducing the degeneration of HBV-infected hepatocytes in a humanized mouse model through Fas/FasL pathway [156]. The FasL expression on NK cells is associated with disease progression of HBV-related acute-on-chronic liver failure [157]. However, recently it has been shown that KLRG1+ NK cells play an antifibrotic role during the natural course of HBV infection. Increased number of KLRG1 positive cells is present in the blood and liver of CHB patients [158]. NK cells kill activated HSC in the liver, which can limit liver scarring. HSCs are the major contributor of hepatic fibrosis and subsequently cause liver dysfunction. Activation of HSCs into proliferative, fibrogenic myofibroblasts is the central driver of hepatic fibrosis in experimental and human liver injury. Therefore, during CHB, NK cells possess both pathogenic as well as protective roles.

### 3.3. CD4 T Cells

CD4 T cells play a critical role in host immunity. They perform various functions including the activation of innate immune cells, B cells, and cytotoxic T cells. Moreover, they promote B cell antibody production, act as APCs, and recruits granulocyte to the site of infection [27]. In HBV infection, CD4 T cells participate in viral clearance by maintaining HBV-specific CD8 T cells [3]. However, they are also involved in the pathogenesis of HBV infection. Antigenic stimulation of CD4 T cells differentiate them into different subtypes including Th1, Th2, Th17, and T_FH_ cells, where Th17 cells play a crucial role in inflammation hepatic fibrosis and HCC development. These cells express chemokine receptor CCR6 and produce IL-17, a pro-inflammatory cytokine that acts through binding to its receptor (IL-17R) present on the surface of various cells. Almost all liver cells including hepatocytes, HSCs, biliary epithelial cells, KCs, and liver sinusoidal endothelial cells express IL-17R [159,160]. Moreover, Th17 cells also secrete IL-22 and granulocyte macrophage colony-stimulating factors. These cytokines encourage neutrophil production by regulating the expression of granulocyte colony stimulating factor and subsequently recruits neutrophil by the regulation CXCR2 ligand, IL-8/CXCL8. Dynamic functions of Th17 cells on different immune cells drive inflammation, tissue damage, and disease progression. Increased frequencies of Th17 cells in HBV patients associate with fibrosis and cirrhosis [161]. Peripheral Th17 cell frequency and serum IL-17 may help in predicting the severity of liver damage and fibrosis. In the liver, IL-17 is mainly localized in the region of hepatic fibrosis and increased IL-17 expression associated with the degree of fibrosis [162]. IL-17 stimulates different chemotactic factors including CXCL8, monocyte chemoattractant ptotein-1 that recruits neutrophils and monocytes leading to marked tissue inflammation. Moreover, it amplifies inflammatory reactions by stimulating different cytokines such as IL-6, IL-1β, TNF-α, and prostaglandin E2. HBV infection causes an imbalance between the ratios of Th17 to Th1 and Th17 to Treg, which are critically involved in disease progression. An increased Th17 to Th1 ratio may promote tumor progression, whereas Th17 to Treg ratio is associated with liver cirrhosis [163,164]. Increased Th17 cells frequencies have been reported in HBV-related acute-on-chronic liver failure (ACLF) non-survivors in comparison to survivor and these frequencies correlate with disease severity indices including CLIF-C score, a mortality predictor. Moreover, Th17 cell frequency could predict 90-day prognosis like MELD score. The cut off value of Th17 cell percentage above 5.9 indicates significantly lower 90-day survival rate in HBV-related ACLF patients, which designates that higher Th17 cells are associated with poor overall survival [165]. Involvement of these cells in disease progression has been described in Figure 2.

Other T helper cell subsets, including Th1 and Th2, are also important players in chronic HBV infection [166]. Naïve T cells differentiate into Th1 cells by early exposure of IL-12 and IFN-γ [167,168]. IFN-γ initiates signaling via signal transducer and activator of transcription 1 pathway and induces T-bet expression, a Th1 transcription factor, which promotes IFN-γ production and downregulates IL-4 expression, a cytokine required for Th2 cells [169]. Upon activation, Th1 cells secrete IFN-γ that induces the activation of DCs and macrophages and encourages their ability to process and present the antigens to T cells [170]. Moreover, Th1 cells secrete TNF-α and IL-2 that provide antiviral defense and promote CD4 and CD8 T cell proliferation and differentiation into effector and memory T cells [171]. On the other hand, Th2 cells facilitate the activation and maintenance of the humoral immune response and produce IL-4, IL-5, IL-10, and IL-13 [172]. Th1 and Th2 mediated immune responses are weaker in CHB patients as compared to healthy volunteers and are associated with the persistence HBV viral load and increased ALT and AST [166]. However, CHB patients with HBeAg+ and high viral load are more strongly associated with the activation of Th1 and Th2 responses than HBeAg- patients. HBeAg was able to downregulate the production of IFN-γ by Th1 cells whereas induced Th2 type cytokines including IL-6 and IL-10 and maintain immune tolerance in CHB patients [173]. Therefore, preferential activation and commitment towards Th1 or Th2 cell subsets may influence the clinical outcomes of HBV infection.

### 3.4. CD8 T Cells

HBV-specific CD8 T cell responses are crucial for viral clearance and are attributed to not only inhibition of viral replication but also apoptosis of infected hepatocytes. However, both HBV-specific as well as global CD8 T cells are also responsible for hepatic inflammation. Resolved HBV patients contain polyclonal and multi-specific CD8 T cell response, while in CHB, CD8 T cells display weak and narrow spectrum of epitopes [174]. CD8 T cells have been studied in the murine model of HBV-induced HCC. HBsAg specific CTLs continuously attack HBsAg-expressing hepatocytes and trigger HCC in HBV transgenic mice. Use of anti-FasL neutralizing antibodies could attenuate HBsAg-specific CTLs hepatotoxicity to prevent chronic hepatitis and eventually HCC [175]. Moreover, increased TGIT expression on hepatic CD8 T cells induces chronic hepatitis and fibrosis and HBV transgenic mice [72]. PD-1+TIGIT+CD8 T cell population is correlated with disease progression as well as poor outcomes in HBV-related HCC, suggesting its importance in clinical implications for prognosis [176]. In addition, stimulation of antigen non-specific memory CD8+ T cells with anti-CD137 mAb leads to the production of IFN-γ in HBV transgenic mice and plays a significant role in the development of chronic inflammation, fibrosis, cirrhosis, and even HCC progression by recruiting hepatic macrophages. These macrophages promoted HCC development through the secretion of IL-6, MCP-1 and TNF-α [151]. CD8+ T cells and CD68+ macrophages could be used as immunological determinants for HBV-related HCC prognosis. Patients exhibiting higher CD68/CD8 ratio presented poor overall as well as disease free survival than those with lower ratio [177]. Expansion of apoptosis associated epitopes specific CD8+ T cells are linked to hepatic fibrosis in CHB [178]. These cells exhibited distinct differentiation states and participated in immunopathology. To summarize, both innate and adaptive immune cells are critical in disease progression and development of fibrosis, cirrhosis, and further HCC.

## 4. Conclusions

Recent knowledge of immune mediated pathologies and potential contribution of innate and adaptive immune cells has been growing rapidly. Chronic HBV infection alters circulating and hepatic microenvironment including substantial modification in innate and adaptive immune cells. Massive phenotypic and functional alterations in the immune cells induce hepatic inflammation and stimulate hepatocyte death in chronic HBV infection. Both innate and adaptive immune cells either work independent or synergistically to drive immunopathogenesis and are equally responsible for disease progression towards fibrosis, cirrhosis, and HCC by producing inflammatory and fibrogenic mediators. Although, nucleos(t)ide analogues are efficient in decreasing viral load and subsequently control disease severity, further understanding of complex interplay between HBV and host immunity is critical to determine whether HBV therapeutic strategies should target the virus or host immunity or both in combination to achieve functional cure. Moreover, interaction between HBV and innate immune cells is instrumental to design novel immunotherapeutics based on the activation and effector function of innate immune cells. Approaches to control excessive inflammation during CHB are also warranted.

## Figures and Tables

**Figure 1 ijms-22-05497-f001:**
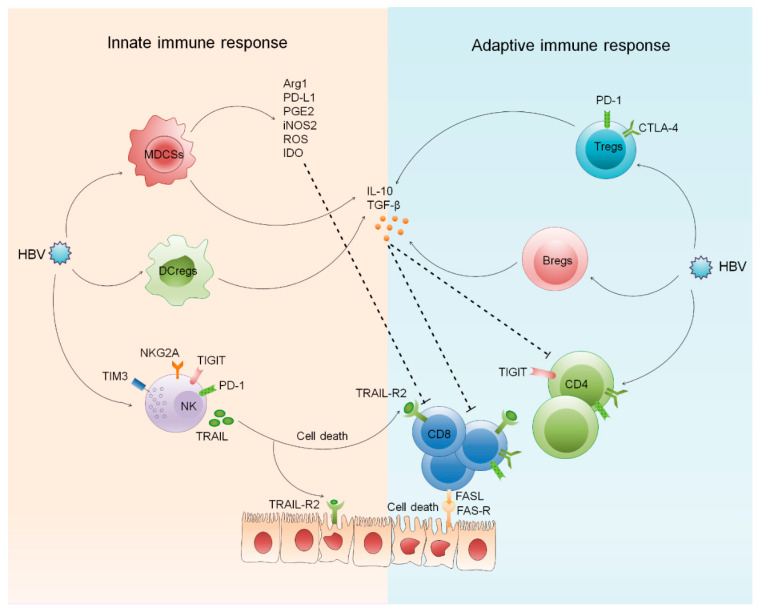
Innate and adaptive immune system derives pathology in CHB. Chronic HBV infection activates different cellular pathways including the innate and adaptive immune system leading to inflammatory cytokine secretion that aggravate inflammation and hepatic injury. To control the excessive inflammation, expansion of regulatory cells comprising, DCregs, MDSCs, Tregs, and Bregs takes place that produce anti-inflammatory cytokines and other components that cause impaired effector function and immune tolerance. Moreover, both innate and adaptive immune cells over express inhibitory receptors and apoptosis inducing receptors and ligands driving immune exhaustion and cell death. HBV: Hepatitis B virus, DCregs: Regulatory dendritic cells, MDSCs: Myeloid derived suppressor cells, Tregs: Regulatory T cells, NK cells: Natural killer cells, TIM-3: T cell immunoglobulin and mucin-domain containing-3, TIGIT: T cell immunoreceptor with Ig and ITIM domain,PD-1: Programmed death-1, TRAIL: Tumor necrosis factor-related apoptosis-inducing ligand, FASL: FAS ligand, FAS-R: FAS receptor, CTLA-4: Cytotoxic T-lymphocyte antigen 4, Arg1: Arginase-1, PD-L1: Programmed death-ligand 1, PGE2: Prostaglandin E2, iNOS: Inducible nitric oxide synthase, ROS: Reactive oxygen species, IDO: Indoleamine-2,3-dioxygenase.

**Figure 2 ijms-22-05497-f002:**
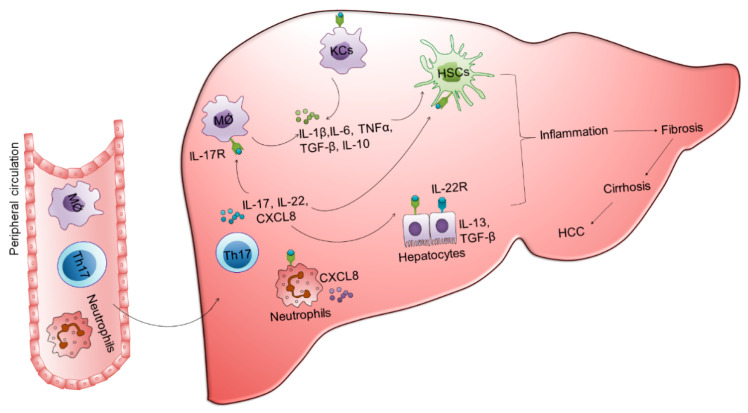
Innate and adaptive immune cells are involved in hepatic inflammation, fibrosis, cirrhosis, and HCC in CHB. Continuous viral exposure prompts the activation of intrahepatic immune cells that initiates the recruitment of circulating immunocytes to the liver. These infiltrated cells stimulate other parenchymal and non-parenchymal cells in the liver. Th17 cells secrete enormous amounts of IL-17 that bind to its receptor present on the surface of many intrahepatic cells and worsen hepatic inflammation by triggering the inflammatory cascade. Induction of profibrogenic mediators IL-10 and TGF-β encourage liver fibrosis by activating hepatic stellate cells. Moreover, continuous supplementation of these fibrogenic stimuli promotes further disease progression towards fibrosis, cirrhosis, and HCC. MǾ: Macrophage, Th17: T helper 17, KCs: Kupffer cells, HSCs: Hepatic stellate cells, TNF-α: Tumor necrosis factor-α, TGF-β: Transforming growth factor- β, CXCL8: Chemokine ligand 8, HCC: Hepatocellular carcinoma.

**Table 1 ijms-22-05497-t001:** Role of inflammatory mediators in CHB.

Cytokines	Functions in CHB Patients	References
IL-6	Mediate HBV entry into hepatocytes, induce inflammation and inflammation-driven HCC by the downregulation of miR-122, inhibits HBV replication, inhibits HBV entry through downregulation of HBV-specific receptor Na(+)/taurocholate cotransporting polypeptide (NTCP)	[97,98,99,100,101]
IL-8	Cause resistance to IFN-α therapy, induce inflammation and apoptosis, induce fibrosis, recruits neutrophil	[102,103]
IL-9	Induce inflammation, necrosis, and fibrosis	[104]
IL-10	Inhibit cytokine production, regulate T cell immunity, develop immune tolerance, and persistence HBV infection	[29,96,105]
IL-12	Reverse immune tolerance towards HBV, induce HBV-specific T cell response, reverse mitochondrial defects in HBV-specific CD8 T cells	[106,107]
IL-13	Induce liver fibrosis and cirrhosis	[108]
IL-15	Enhance CD8 T cell response, upregulate PD-1 and PD-L1	[109]
IL-17	Immune activation, inflammation, induce liver fibrosis	[110,111,112]
IL-18	Increase risk of cirrhosis, enhance IFN-γ release, and improves clearance of virus infected cells	[113]
IL-21	Activate T and B cells, induce IFN-γ secretion and clear HBV antigen, generation of plasmablasts and plasma cells, development of HBV induced liver cirrhosis, and exacerbates liver injury	[20,114]
IL-22	Inhibit liver inflammation and fibrosis, induce fibrosis and HCC	[115,116]
IL-23	Induce inflammation and HCC development	[117]
IL-27	Support plasmablasts and plasma cell generation, enhance HBsAg-specific antibody production, inhibits HBV protein expression and viral capsid associated DNA replication	[20,118,119]
IL-33	Induce liver damage and fibrosis, Activate T_FH_ cells and enhance humoral immunity, suppress HBV replication and HBeAg secretion	[120,121]
IL-35	Development of cirrhosis and HCC, inhibit HBV-specific CD8 T cells proliferation and cytotoxicity, inhibit cytokine induce antiviral immunity	[122,123]
IFN-γ	Antiviral immunity, promote CXCL-9 secretion by macrophages, inhibit HBV replication, induce inflammation	[124,125,126]
TNF-α	Inhibit HBV replication, provide antiviral immunity, induce inflammation	[127,128]
TGF-β	Impair NK cell function, encourage fibrosis and HCC	[129,130]

## Data Availability

Not applicable.

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
