# Peer review of "Immunopathology of Chronic Hepatitis B Infection: Role of Innate and Adaptive Immune Response in Disease Progression"

_ijms, 2021, doi:10.3390/ijms22115497_

Round 1

Reviewer 1 Report

In this review of immunopathology during chronic HBV infection the authors provide a broad perspective of the many immunocytes that participate in inflammation, cirrhosis, viral control and HCC. Focus of the discussion is on the immunopathogenic effects of cell types, function, receptors and cytokines elaborated. DC, MDSC, Treg, Breg, CD4, CD8, monocytes, macrophages, NK cells are covered, as are inhibitory receptors. A nice model connecting a number of these players, and a summary table of the contribution of different cytokines is included.

Comments

This is a very nice overview/summary that will be informative to all.

It seems that since the role of T cells in control of HBV is covered it would also be appropriate to include discussion about the role of B cells in control of occult HBV, disrupted through use of rituximab, tells us about the role of B cells in control of HBV.

Author Response

We sincerely appreciate the reviewer’s comment. As per the reviewer’s suggestion, we have included the role of B cells in controlling HBV infection and also discussed about rituximab therapy in increasing the risk of HBV reactivation under the heading “Involvement of B Cells in CHB” ( Point number: 2.3, pages: 8-9, lines: 349-412) and highlighted it with red color. We have added the references accordingly.

Reviewer 2 Report

This review from Khannam and colleagues addresses the roles of the innate and adaptive components of the immune system in the immunopathology of chronic HBV. As HBV is a major global health concern and immunopathology is a major component of disease progression. The authors appear to have done a rigorous job of addressing relevant areas in this field and the table and figure that are present are helpful.

Sadly in its current state the manuscript is not suitable for publication. The main problem is the quality of the English, with errors throughout. The manuscript would benefit from the input of a language editor. Additional figures would also help clarify some of the concepts discussed.

Below I’ve included a few examples of phrases that require improvement, but it is in no way a comprehensive list.

Line 190-1 - “Normally, Tregs relieve liver inflammation and immune mediated liver injury; however, they may also regulate apoptosis-induced inflammation.” Relieve implies reduced inflammation, but regulate could mean increase or decrease.

Line 197 - “Increased Tregs in the liver tissues of CHB patients with severe hepatitis and suggested that increased Tregs associated with chronicity and degree of inflammation [46].” Does not seem to be a complete sentence.

Line 215 - “CD19+CD24hiCD38hi transitional B cells, CD19+CD24hiCD27+ B cells” the significance of these cell types is not explained.

Line 264-70 - references?

Line 270 - “During acute HBV infection, effector T cells retain higher 270 PD-1 expression that tend to decline in the recovery phase [62].” Relevance of PD-1 is not explained.

Line 288 - “Recently, it has been reported that 288 CTLA-4 expression on Tregs inhibit the function of T follicular helper (TFH) cells and its 289 blockade with CTLA-4 neutralizing antibody restore the ability of TFH cells to clear the 290 infection [27].” The particular relevance of follicular T cells is not explained.

Line 319 - atMBCs are not defined.

Line 337 - “Cytokines are required for cellular communication, their activation, and provide appropriate defense against different pathogens.” Should be re-written.

Table 1 - IL-6 cited as promoting HBV entry, ignoring several papers suggesting that IL-6 reduces HBV infection by downregulating hNTCP expression.

Line 421 - “CHB infection” CHB is a disease, it doesn’t infect anything.

Line 464 - This sort of basic information about the role of CD4 T cells

Line 488 - It is unclear specifically what the Th17 cell percentage refers to.

Line 515 - The paper introduces the idea of a discussion of the relative contribution of adaptive and innate immunity to HBV immunopathogenesis, but the summary simply states that both are involved. Either a more decisive summary or a less click-bait title might be appropriate.

Author Response

  1. We are thankful to the reviewer for deep and thorough review of the manuscript. As per the reviewer’s kind suggestion, we have taken the inputs of a language editor and revised the manuscript accordingly. The changes are highlighted in red color. Moreover, we have included additional figure (figure 2) as suggested by the reviewer, to clarify few of the mechanisms discussed in the article.
  2. We appreciate the reviewer’s comment.  As per the reviewer suggestion, we have revised the sentence.

  3.  

    We thank reviewer for the comment. As per the reviewer suggestion, we have revised and completed the sentence.

  4.  

    We appreciate the reviewer’s comment. As per the reviewer concern, we have included the significance of both CD19+CD24hiCD38hi transitional B cells and CD19+CD24hiCD27+ B cells and added the references accordingly.   

  5.  

    As per the reviewer suggestion, we have included the references in the respective paragraph. 

  6.  

    We are thankful to the reviewer for the comment; we have included the relevance of PD-1 and revised the sentence.

  7.  

    As per the reviewer’s suggestion, we have included the function and relevance of follicular T cells and added the references as needed.

  8.  

    As per the reviewer’s suggestion, now we have defined atMBCs.

  9.  

    As per the reviewer’s concern, we have revised the sentence.

  10.  

    As per the reviewer’s suggestion, we have included the role of IL-6 in reducing HBV infection through downregulation of hNTCP in table 1 and added the references.

  11.  

    As suggested by the reviewer, we have replaced CHB infection with “chronic HBV infection” throughout the manuscript.  

  12.  

    As per the reviewer concern, we elaborated about the role of CD4 cells in CHB and revised the paragraph. Moreover, we included a paragraph on the role of Th1 and Th2 cells during chronic HBV infection under the heading “CD4 T cells”. 

  13.  

    We expanded about the role of Th17 cells during chronic HBV and specifically discussed what Th17 cell percentage refers to.

  14.  

    As per the reviewer’s suggestion, we have revised the summary.

    We have highlighted all the changes in the manuscript  with red color. 

Reviewer 3 Report

It is an interesting review about “Immunopathology of Chronic Hepatitis B Infection”.

CHB shows a defective early innate immune response, which are essential for the further induction of HBV-specific adaptive immunity and may contribute to the persistence of CHB or a weakened capacity to clear HBV. Viral loads were previously shown to affect the quality of the anti-HBV immune responses and outcomes of viral infections [36]. The actual elimination of HBV infection requires the presence of adaptive immune responses. On the other hand, upon exposure to high-level HBV, human macrophage could be activated with increased inflammatory cytokine expression. However, by quantification of serum cytokines, a study, which enrolled 21 HBV-infected patients during the pre-symptomatic phase, indicated that HBV infection did not elicit production of IFNs and IL-15, but did induce the production of IL-10. Changes in the balance of cytokine profiles may result in either persistence of HBV infections. CHB with HBeAg (+) with high viral loads is more strongly associated with the activation of Th1-and Th2-type responses than CHB with HBeAg (-). Thus, preferential activation and commitment towards Th1- or Th2-cell subsets may influence the clinical consequences of HBV infection.

The study of the interaction between HBV and innate immune cells is instrumental to design novel immune-therapeutic concepts based on the activation/restoration of innate cell functions and/or innate effectors.

Above mentioned should be referred to.

Author Response

We are thankful to the reviewer for the comment. We have revised the manuscript and included the reference suggested by the reviewer. We have also incorporated a paragraph on the role of Th1 and Th2 cells under the heading “CD4 T Cells” (point number 3.3). All the changes are highlighted with red color.

Round 2

Reviewer 2 Report

In this revised manuscript the authors have addressed several of the issues raised in my first review and several elements (cell types, surface markers) that I felt were previously poorly described have been improved.

The writing could still be improved, but the science should be mostly intelligible, albeit primarily for an immunologist rather than virologist audience. 

That said, a brief description of the HBV proteins, near the beginning of the review would be helpful to those not intimately familiar with HBV biology.HBc, HBsAg, etc are frequently mentioned as targets of various immune responses, but their significance to the virus is not explained. For example, knowing which proteins are exposed on the cell surface, the virion surface, which are secreted, is significant to concept such as virus neutralisation, immune tolerance, cell exhaustion, etc.

Specific points (as before, this is not an exhaustive list)

Fig 1 legend - Defined “Dregs”? DCregs?

Line 315 define T-bet/Tbet mRNA, what is it, why does it matter?

Line 352 - antibodies against particles or proteins?

Line 466 - define THP-1 cells?

Define HSC(s)

I still don’t like the fact that the title asks, "Who is the Bigger Culprit Innate or Adaptive Immunity?”, then concludes, “To summarize, both innate and adaptive immune cells are critical in disease progression and development of fibrosis, cirrhosis and further HCC.” The review simply describes the effects of innate and adaptive immunity. I accept that “both are important” is the reality, particularly given extensive cross-talk, but I don’t think the title should pose a question that the article makes no attempt to resolve. 

Author Response

  1. We are thankful to the reviewer for the comment. As per the reviewer’s suggestion, we included a brief paragraph on different HBV proteins and their functions under the heading  "Innate and Adaptive Immune Response Against HBV infection" on page 2, lines 90-103. 
  2. We thank reviewer for the comment. It was a typographical error. We have replaced Dregs with DCregs.  

  3. We appreciate the reviewer’s comment. As suggested by the reviewer, we defined T-bet/T-bet mRNA expression and described its relevance.

  4. It is protein. As per the reviewer’s concern, we replaced particles with proteins. 
  5. We have defined THP-1 cells.

  6. We have defined HSCs.

  7. We thank reviewer for the comment. As per the reviewer’s concern, we have revised the title of the manuscript. The current title is “Immunopathology of Chronic Hepatitis B Infection: Role of Innate and Adaptive Immune Response in Disease Progression”. We hope that the reviewer will find the title appropriate as per the content presented in the manuscript.

         We have highlighted all the changes with red color.